# GASPACHO: G̲aussian S̲pl̲atting for C̲ontrollable H̲umans and O̲bjects

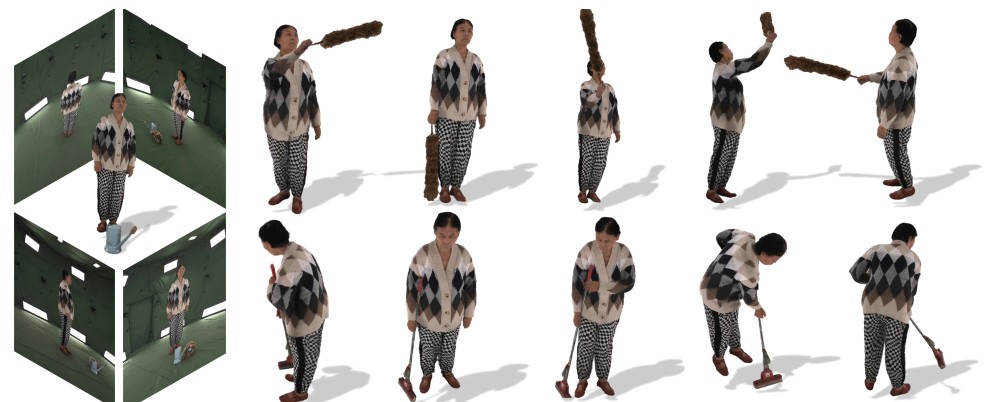

Figure 1. **Left**. Using multi-view RGB images of human-object interactions, our method jointly reconstructs animatable models of the human and the object. **Right** The reconstructed model can be driven by control signals to synthesize photo-real *novel pairs* of human-object interactions with *free camera viewpoint control*.

## ABSTRACT

We present GASPACHO, a method for generating photorealistic, controllable renderings of human–object interactions from multi-view RGB video. Unlike prior work that reconstructs only the human and treats objects as background, GASPACHO simultaneously recovers animatable templates for both the human and the interacting object as distinct sets of Gaussians, thereby allowing for controllable renderings of novel human object interactions in different poses from novel-camera viewpoints. We introduce a novel formulation that learns object Gaussians on an underlying 2D surface manifold rather than in 3D volume, yielding sharper, fine-grained object details for dynamic object reconstruction. We further propose a contact constraint in Gaussian space that regularizes human–object relations and enables natural, physically plausible animation. Across three benchmarks—BEHAVE, NeuralDome, and DNA-Rendering—GASPACHO achieves high-quality reconstructions under heavy occlusion and supports controllable synthesis of novel human–object interactions. We also demonstrate that our method allows for composition of humans and objects in 3D scenes and for the first time showcase that neural rendering can be used for the controllable generation of photoreal humans interacting with dynamic objects in diverse scenes.

## 1 INTRODUCTION

Photorealistic novel view synthesis of human avatars is essential for several applications ranging from telepresence, AR/VR, special effects to e-commerce. The recent emergence of Gaussian Splatting as a 3D representation (Kerbl et al., 2023) has enabled a new wave of efficient methods that reconstruct high-quality animatable human avatars (Moreau et al., 2024; Wen et al., 2024; Kocabas et al., 2023; Qian et al., 2023; Li et al., 2024b; Pang et al., 2023). These articulated human models can be animated by controllable human pose signals to deform the 3D Gaussians into a novel pose and then rendered from any camera viewpoint in real-time. However, these methods assume that humans are *recorded in isolation*, without any occlusions, and when they are animated, their actions are generated in *isolation*.

In this work, we use multi-view (sparse and dense) captures of human interacting with objects and aim to create animatable 3D Gaussians models for both the human and the object. The reconstructed models can be used to render the captured scene from novel viewpoints, but our main goal is to generate novel human-object interactions that can be integrated in 3D scenes and rendered in real-time. While there exists a body of work on human-object interaction (Guzov et al., 2021; Zhang et al., 2022; Hassan et al., 2019; Zhu et al., 2024; Hassan et al., 2021a), it focuses only on modeling human behaviour in environments and *neglects human appearance*. A few recent papers study the photorealistic rendering/reconstruction of human-object interaction, but assume the objects to be static (Kocabas et al., 2023; Xue et al., 2024), focus only on human-object reconstruction and are unable to synthesize *novel human-object interactions* (Sun et al., 2021; Jiang et al., 2022b; Wang et al., 2025a) or assume the object to be a small extension of one human hand (Zhan et al., 2024; Liu et al., 2023). The new problem we define is challenging in several ways:

First, Human-Object interactions inevitably creates significant occlusions of the human body and/or the object. We observe that these occlusions happen to be an important problem for existing human reconstruction pipelines that fail on these occluded regions and propose a composition and occlusion guided loss to address it.

Then, animatable human reconstruction methods are facilitated by pre-computed pose estimation of the SMPL (Loper et al., 2015) body template to initialize and track the human geometry. However, we do not assume that object geometry and tracking is known beforehand. Because objects are often small and occluded, obtaining an accurate template often leads to degenerate solutions with common techniques. Instead, we propose to reconstruct objects from scratch in a coarse-to-fine manner, that includes learning a template in features space, tracking it across frames with photometric alignment and finally use *pose-independent* Gaussian maps to learn high-fidelity textures.

Finally, beyond reconstruction, generating novel interactions from avatars and objects while maintaining a physically plausible contact is not straightforward. Having separated animatable models potentially enables interesting scenarios: reanimate the observed human and object with a novel interaction, replace one human by another to execute the same interaction, or both at the same time. However, determining the correct control parameters for these scenarios is difficult because they depend on the body shape of the human and the geometry of the object. We address this problem by introducing a novel human-object contact constraint in Gaussian space to generate interactions with natural human-object contacts.

We evaluate our method on DNA rendering (Cheng et al., 2023a), NeuralDome (Zhang et al., 2023), and BEHAVE datasets (Bhatnagar et al., 2022), showing significant improvement over existing methods for joint human-object reconstruction. DNA-Rendering provides dense camera setups, while evaluating on BEHAVE demonstrates the utility of our method for data collected with sparse (only four cameras) low-cost mobile, commodity sensors. We also show photorealistic demonstrations of novel interactions between avatars and objects reconstructed from different datasets and integrated in 3D Gaussians scenes. To the best of our knowledge, such demonstrations were not achieved before and are possible thanks to the multiple contributions presented in this paper.

To summarize, our most important contributions are: 1) We present a novel method to jointly reconstruct humans and objects models under occlusion, which can then be animated to synthesize novel human-object interactions. 2) We introduce a coarse-to-fine pipeline for reconstructing *dynamic* objects under interaction-induced occlusion, with features space template, 3D tracking and refinement with *pose-independent* Gaussian maps. 3) We introduce human-object contact constraints in Gaussian space to ensure proper contacts when animating 3DGS humans interacting with novel objects. 4) We demonstrate that our method allows for photorealistic and controllable composition of humans, objects and 3D scenes coming from diverse captures, an application that was not seen before to the best of our knowledge.

## 2 RELATED WORK

**Neural Rendering:** Since the publication of NeRF (Mildenhall et al., 2020), there has been an explosion of interest in Neural Rendering (Xie et al., 2022b). Despite appealing image quality, NeRF is limited by its computational complexity. Though there have been several follow-up improvements (Müller et al., 2022; Barron et al., 2022; 2023; Tancik et al., 2023), the high computational cost of NeRF remains. 3DGS introduced by (Kerbl et al., 2023) addresses this limitation by repre-

senting scenes with an explicit set of primitives shaped as 3D Gaussians, extending previous work using spheres (Lassner & Zollhöfer, 2021). 3DGS rasterizes Gaussian primitives into images using a splatting algorithm (Westover, 1992). 3DGS originally designed for static scenes has been extended to dynamic scenes (Shaw et al., 2023; Luiten et al., 2024; Wu et al., 2024; Lee et al., 2024; Li et al., 2023a), slam-based reconstruction, (Keetha et al., 2024), mesh reconstruction (Huang et al., 2024; Guédon & Lepetit, 2024) and NVS from sparse cameras (Mihajlovic et al., 2024) and embodied views (Wang et al., 2025b).

**Human Reconstruction and Neural Rendering:** Mesh-based templates (Pavlakos et al., 2019; Loper et al., 2015) have been used to recover 3D human shape and pose from images and video (Bogo et al., 2016; Kanazawa et al., 2018). However, despite recent efforts and improvements (Alldieck et al., 2018; 2019; Shin et al., 2024), mesh-based representation does not often allow for photorealistic renderings. Implicit functions (Mescheder et al., 2019; Park et al., 2019) have also been utilized to reconstruct detailed 3D clothed humans (Chen et al., 2021; Alldieck et al., 2021; Saito et al., 2020; He et al., 2021; Huang et al., 2020; Deng et al., 2020). However, they are also unable to generate photorealistic renderings and are often not reposable. Several works (Peng et al., 2021; Guo et al., 2023; Weng et al., 2022; Jiang et al., 2022a; Habermann et al., 2023; Zhu et al., 2024; Li et al., 2022; Liu et al., 2021; Xu et al., 2021) build a controllable NeRF that produces photorealistic images of humans from input videos. However they inherit the speed and quality restrictions of the original NeRF formulation. Furthermore, unlike us, they do not model human-object interactions. With the advent of 3DGS, several recent papers use a 3DGS formulation (Kocabas et al., 2023; Qian et al., 2024; Moreau et al., 2024; Abdal et al., 2024; Zielonka et al., 2023; Moon et al., 2024; Li et al., 2024b; Pang et al., 2024; Lei et al., 2023; Hu et al., 2024; Li et al., 2024a; Zheng et al., 2024; Jiang et al., 2024b; Dhamo et al., 2024; Qian et al., 2023; Xu et al., 2024; Guo et al., 2025; Qiu et al., 2025; Junkawitsch et al., 2025; Moreau et al., 2025) to build controllable human or face avatars. Unlike our method, they do not model human-object interactions or reconstruct humans under occlusion. Prior works have also extended the 3DGS formulation to model humans along with their environment, (Xue et al., 2024; Zhan et al., 2024), but unlike us, they either assume that the 3D scene remains static or that its constituent parts are an extension of the human hand.

**Human Object Interaction:** Human Object Interaction is another recurrent topic of study in computer vision and graphics. Early works (Fouhey et al., 2014; Wang et al., 2017; Gupta et al., 2011) model affordances and human-object interactions using monocular RGB. The collection of several recent human-object interaction datasets (Hassan et al., 2021a; Guzov et al., 2021; Hassan et al., 2019; Savva et al., 2016; Taheri et al., 2020; Bhatnagar et al., 2022; Jiang et al., 2024a; Cheng et al., 2023b; Zhang et al., 2022) has allowed the computer vision community to make significant progress in joint 3D reconstruction of human-object interactions (Xie et al., 2022a; 2023; 2024; Zhang et al., 2020). These datasets have also led to the development of methods that synthesize object conditioned controllable human motion (Zhang et al., 2022; Starke et al., 2019; Hassan et al., 2021b; Diller & Dai, 2024). All these methods represent humans and objects as 3D meshes and as such inherit all the limitations of mesh-based representations including their inability to generate photorealistic images, while our method allows for photorealistic renderings of humans and objects.

## 3 METHOD

Our method receives a set of multi-view RGB images of humans interacting with objects, captured from $N$ cameras at $T$ temporal frames, $\{\mathbf{I}_t^c\}_{t=1,c=1}^{t=T,c=N}$, along with estimates of camera parameters $\{\gamma_t^c\}_{t=1,c=1}^{t=T,c=N}$ and human pose $\{\theta_t\}_{t=1}^T$. We aim to learn a deformable function $f(\theta, \phi; \gamma)$ that maps new human poses $\theta$ and object poses $\phi$ to RGB images rendered from novel viewpoints $\gamma$. **Intuitively**, we are interested in a function that allows us to map human and object poses to 3D Gaussians, which can be used to render novel human-object interactions with free camera viewpoint control.

We first use a feature-based representation to construct coarse object templates (Sec. 3.1) and track the template across the temporal frames (Sec. 3.2) to obtain estimates of object pose parameters. Using the estimated object template and the SMPL mesh we define 2D canonical human and object Gaussian maps to learn Gaussian properties (Sec. 3.3) of two sets of Gaussians $\mathcal{G}_O$ and $\mathcal{G}_H$ anchored to the canonical object and SMPL templates respectively. These are deformed using the provided human and estimated object pose parameters to posed space (Sec. 3.4) and learnt using an occlusion

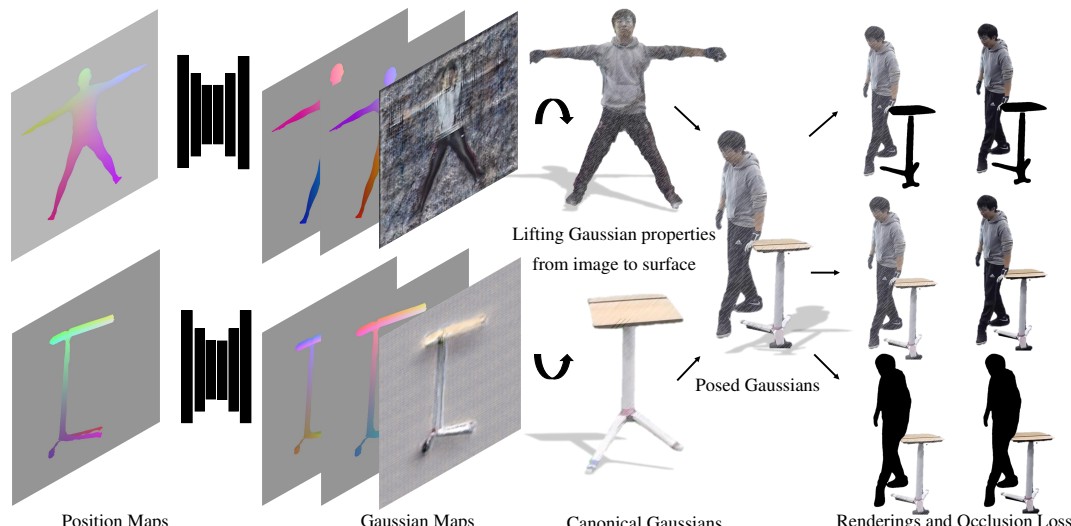

Figure 2: **Method Overview:** Using position maps as input, we learn Gaussian parameters for Gaussians anchored to canonical human and object templates. Gaussian properties - orientation, scale, opacity, color - are learnt as 2D maps structured according to a UV unwrapping of canonical human and object templates. Each pixel thus corresponds to a Gaussian in the canonical template. Once mapped to canonical Gaussians, these Gaussians are posed using LBS for the human and a rigid transform for the object. We render the posed object and human Gaussians separately and compare against the segmented portions of the target image. We further guide the reconstruction using known occlusion information. In the figure above, the black regions of the rendered images are masked out thereby allowing our method to deal with occlusions. In the 2D maps, the gray regions indicate pixels which don't map to any 3D Gaussians.

guided photometric loss (Sec. 3.5). Finally at test time, we introduce a human-object contact (Sec. 3.6) constraints when humans are animated interacting with novel objects.

## 3.1 CANONICAL OBJECT TEMPLATE

To model object motion, we first learn a canonicalised 3D object template. Direct optimization of 3DGS Kerbl et al. (2023) fails under naturally occurring human-object occlusions (Tab. 3), so we adopt feature-based planes Mihajlovic et al. (2024). The features act as a regulizer and smoothness prior which allows for better convergence under occlusion.

For feature-based object Gaussians reconstruction, we learn 3 - $M \times W$, $l$-dimensional planes $\mathbf{F} \in \mathbb{R}^{3 \times M \times W \times l}$ - where $l$ is the feature dimension, for the object. Each Gaussian location for the object $\mathbf{x}_p$ (where $p$ indexes the Gaussian in the set of Object Gaussians) is projected onto the three feature planes to obtain feature vectors - using standard orthographic projection. This idea is analogous to the feature based representation introduced in Chan et al. (2021) and also used in Kocabas et al. (2023), but we show its utility for template reconstruction of occluded objects. These features are then concatenated along the feature dimension. We denote this operation - that maps object Gaussian location $\mathbf{x}_p$ to its corresponding sampled feature $\mathbf{f}_p$ - as $\mathbf{f}_p = \pi(\mathbf{x}_p; \mathbf{F}) \in \mathbb{R}^{3l}$. Using multiple small MLPs we map these feature vectors onto Gaussian parameters - color, covariance, scale, opacity. We learn the positions of the Gaussians directly; query the features at these positions and map these queried features to the rest of the Gaussian parameters. Hence, though we represent the canonical Gaussians using standard 3DGS parameters, we do not learn these parameters directly. Instead, we learn the network weights, feature grids and Gaussian locations $\mathbf{x}_p$ that are mapped onto these 3DGS parameters. Once the features are learnt, they can be discarded after querying them at gaussian positions to obtain other parameters. For an illustration of occlusion and feature based learning, we refer to the supplementary.

We pick a frame where the object is minimally occluded. To do this, we compute the number of object pixels (using object segmentation masks across all cameras) in the images at timestep $t$; of all the temporal frames, the frame with the most number of object pixels is considered to be the one with minimal occlusion. Using this frame $t^*$ with minimal occlusion, we optimize $\mathcal{G}_O^* = \arg\min_{\mathcal{G}_O} \sum_{c=1}^{N} \|\mathcal{R}(\mathcal{G}_O; \gamma_c) - \mathbf{O}_{t^*}^c \cdot \mathbf{I}_{t^*}^c\|_2$, where $\mathcal{R}(.; \gamma)$ is a 3DGS rasterizer with camera parameters $\gamma$, $\mathbf{O}_t^c$ the object mask, and $\mathbf{I}_t^c$ the RGB image, i.e we optimize the 3DGS parameters

which match the segmented object pixels in one set of multiview images. This setup closely matches standard 3DGS reconstruction but with a feature based parameterization and segmented object images as targets. We want to highlight that we use the feature based parameterization detailed above - i.e we do learn the Gaussian properties - just not directly but parameterized by Gaussian positions and the features. This yields a coarse template $\mathcal{G}_O^*$.

## 3.2 OBJECT TRACKING

Once we have learnt a set of 3D Gaussians describing one frame $\mathcal{G}_O^*$, we want to optimize the transformations that map this canonical object to the renderings in subsequent timesteps $t$. Thus, we apply an optimizable rigid 6D pose transform $\phi$ to the object template $\mathcal{G}_O^*$, and minimize the per-pixel error between renderings and target images

$$\arg\min_{\phi_t} \sum_{c=1}^{N} ||\mathcal{R}(\mathcal{T}(\mathcal{G}_O^*; \phi_t); \gamma_c) - \mathbf{O}_t^c \cdot \mathbf{I}_t^c||_2 \ . \tag{1}$$

Here we define $\mathcal{T}(\mathcal{G}_O; \phi)$ to be a rigid transformation on the position and covariance of canonical object Gaussians. Specifically, for every canonical 3D Gaussian in the set $\mathcal{G}_O^*$, we transform its position $\mathbf{x}$ and covariance $\mathbf{\Sigma}$ attributes:

$$\mathbf{x}' = \mathbf{R}_\phi \mathbf{x} + \mathbf{t}_\phi, \quad \mathbf{\Sigma}' = \mathbf{R}_\phi \mathbf{\Sigma} \mathbf{R}_\phi^\top, \tag{2}$$

where $\mathbf{R}_\phi$ and $\mathbf{t}_\phi$ are the rotation and translation components of the 6D object pose $\phi$. To ensure convergence, $\phi_t$ is initialized from $\phi_{t-1}$.

## 3.3 GAUSSIAN MAPS

We use the SMPL body model to first define a canonical human Gaussian template $\mathcal{G}_H^*$ by densely sampling points on the SMPL mesh and setting initial color to zero, opacity and covariance to a fixed constant. Given the canonical templates $\mathcal{G}_H^*$ and $\mathcal{G}_O^*$, we now learn their Gaussian properties (position offsets, covariance, opacity, color) in image space using two StyleUNet models $S_H$ for the human and $S_O$ for the object. This 2D formulation leverages strong inductive biases of CNNs (Pang et al., 2024; Li et al., 2024b; Hu et al., 2023) while maintaining a one-to-one correspondence between pixels and 3D Gaussians.

**Human maps (pose-dependent):** For the human we project the SMPL body model in canonical pose from two front and back views (similiar to Fig. 3) to obtain canonical UV maps. Each pixel in the 2D map is assigned to a canonical 3D human Gaussian in $\mathcal{G}_H^*$. At timestep $t$ with pose $\theta_t$, to provide pose and shape information to a 2D CNN (following (Li et al., 2023b)), we first create pose-dependent position maps for the human by using LBS to deform the canonical SMPL mesh vertices without any global orientation or translation. Each deformed vertex is stored in the corresponding location of the UV map to define a posed position map (similiar to Fig. 3). A StyleUNet (Wang et al., 2023) $S_H$ takes these maps as input (Fig. 2) and predicts per-pixel Gaussian properties: opacity, color, position, covariance as offsets from canonical human template Gaussian.

**Object maps (pose-independent):** For the object, we project the coarse canonical template $\mathcal{G}_O^*$ from front and back views (Fig. 3) to define mappings between a 2D image and 3D object Gaussians. To define fixed pose-independent position map we directly store the location of the Gaussian in the template at the 2D projected pixel location (Fig. 3). Unlike humans, this map does not vary with time and is reused at every timestep. To our knowledge, such pose-independent Gaussian maps for rigid objects are novel, and we find that they stabilize 6D pose optimization and yield more consistent renderings. A second StyleUNet $S_O$ processes these maps and predicts offsets of Gaussian properties from the canonical template.

## 3.4 DEFORMATION

The canonical Gaussians $\mathcal{G}_H^*$ and $\mathcal{G}_O^*$ are first updated by the StyleUNets (Sec. 3.3). For both humans and object, $S_H$ and $S_O$ predict offsets from the canonical templates: position, covariance, opacities, color; additionally $S_H$ also predicts per-gaussian (human Gaussians) skinning weights $\mathbf{w}_k^H$. We use

Figure 3: Object Gaussian Maps. Each pixel corresponds to a canonical Gaussian. Object maps are pose-independent, a novel design that yields high-fidelity textures. Arrows indicate correspondence. To define position-maps we store the 3D location of Object Gaussians corresponding to a pixel in the 2D map.

these offsets to update canonical Gaussians and refer to these updated Gaussians as $\mathcal{G}_H$ and $\mathcal{G}_O$. In practice the human canonical Gaussian change per frame.

We then deform these StyleUNet-modified Gaussians into posed space. For humans, Linear Blend Skinning (LBS) (Fig. 2) is applied using the learned weights $\mathbf{w}_k^H$, giving per-Gaussian transformations $\mathbf{R}_k^\theta = \sum_{j=1}^J \mathbf{w}_{kj}^H \mathbf{R}_j^{SMPL}(\theta)$. Here we use $\mathbf{R}_j^{SMPL}(\theta)$ to denote the $j^{th}$ joint-transformation matrix of the SMPL body pose - we use $k$ to denote the $k^{th}$ human Gaussian.

For every canonical 3D human Gaussian in the set of canonical Gaussians $\mathcal{G}_H$, we transform its position $\mathbf{x}_k$ and covariance $\mathbf{\Sigma}_k$ attributes as:

$$\mathbf{x}_k' = \mathbf{R}_k^\theta \mathbf{x}_k + \mathbf{t}_k^\theta, \quad \mathbf{\Sigma}_k' = \mathbf{R}_k^\theta \mathbf{\Sigma}_k \mathbf{R}_k^{\theta\top}, \tag{3}$$

Here we use $\mathbf{R}_k^\theta$ and $\mathbf{t}_k^\theta$ to denote the rotational and translational components of the per-gaussian transformation matrix. For objects, rigidity is assumed and a global 6D transform with pose $\phi$ is applied - using $\mathcal{T}(\mathcal{G}_O; \phi)$ (defined in Sec. 3.2). This process yields posed human and object Gaussians (Fig. 2). In both cases - object and human - only positions and covariances are deformed, while colors and opacities remain as predicted by the StyleUNets.

## 3.5 COMPOSITION AND OCCLUSION GUIDED LOSS

Given input images, $\mathbf{I}_t^c$ with human and object masks $\mathbf{H}_t^c, \mathbf{O}_t^c$, and human and object poses: $\theta_t$ and $\phi_t$ - with camera index $c$, timestep $t$ - we render posed human and object Gaussians $\mathcal{G}_H^t = \mathcal{T}(\mathcal{G}_H; \theta_t)$ and $\mathcal{G}_O^t = \mathcal{T}(\mathcal{G}_O; \phi_t)$ to obtain $\mathbf{I}_H^{ct} = \mathcal{R}(\mathcal{G}_H^t; \gamma_c)$, $\mathbf{I}_O^{ct} = \mathcal{R}(\mathcal{G}_O^t; \gamma_c)$, and $\mathbf{I}_A^{ct} = \mathcal{R}(\mathcal{G}_H^t \cup \mathcal{G}_O^t; \gamma_c)$. We use the input images and segmentation masks to supervise these renderings.

To avoid penalizing occluded regions, we ignore pixels where the other entity is visible. The per-frame, per-camera losses are $\mathcal{L}_H^{ct} = \|\mathbf{I}_t^c \mathbf{H}_t^c (1 - \mathbf{O}_t^c) - \mathbf{I}_H^{ct}(1 - \mathbf{O}_t^c)\|_1$, $\mathcal{L}_O^{ct} = \|\mathbf{I}_t^c \mathbf{O}_t^c(1 - \mathbf{H}_t^c) - \mathbf{I}_O^{ct}(1 - \mathbf{H}_t^c)\|_1$, and $\mathcal{L}_A^{ct} = \|\mathbf{I}_t^c(\mathbf{H}_t^c + \mathbf{O}_t^c) - \mathbf{I}_A^{ct}\|_1$.

The overall image loss is $\mathcal{L}_1 = \sum_{c,t}(\mathcal{L}_H^{ct} + \mathcal{L}_O^{ct} + \mathcal{L}_A^{ct})$. We additionally use a perceptual loss $\mathcal{L}_{per}$ (Zhang et al., 2018) with the same masking strategy and a regularization loss $\mathcal{L}_{reg}$ that encourages predicted skinning weights to remain close to SMPL-derived weights. The final objective is $\mathcal{L} = \lambda_{L1}\mathcal{L}_1 + \lambda_{per}\mathcal{L}_{per} + \lambda_{reg}\mathcal{L}_{reg}$.

This formulation prevents erroneous supervision in occluded regions, while multi-view consistency ensures that ignored Gaussians are still supervised in other frames or views. Furthermore our *pose-independent* Gaussian map formulation allows us to optimize the 6D object pose during training as well and hence refine the initially tracked object 6D parameters.

## 3.6 GAUSSIAN HUMAN-OBJECT CONTACT REFINEMENT

A key novelty of our framework is that it enables reanimating a reconstructed human identity with novel motions and novel objects. Given new motion parameters $\theta_t$ and $\phi_t$, the human StyleUNet $S_H$ and object StyleUNet $S_O$ are used to predict canonical Gaussian $\mathcal{G}_O$ and $\mathcal{G}_H$ These are deformed with LBS for humans and a rigid transform for objects to obtain posed Gaussians $\mathcal{G}_H^t = \mathcal{T}(\mathcal{G}_H; \theta_t)$ and $\mathcal{G}_O^t = \mathcal{T}(\mathcal{G}_O; \phi_t)$. The naive composition of these two sets of Gaussians often causes penetrations or missing human-object contacts, so we further regularize these two sets of Gaussians by optimizing displacements $\mathbf{\Delta}$ for human contact Gaussians. However unlike SMPL, Gaussians Avatars

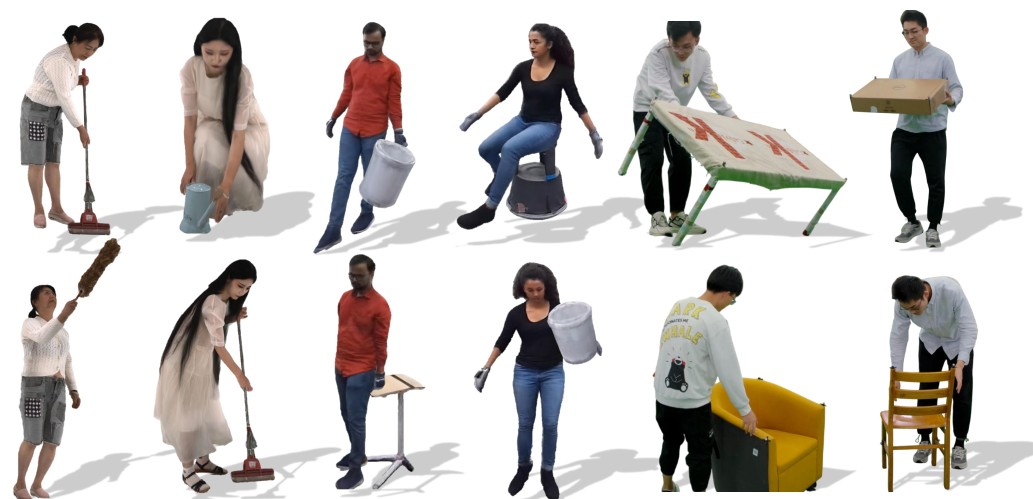

Figure 4: **Top Row:** Qualitative Results of our method on the DNA-Rendering (left), BEHAVE (middle) and Neural Dome (right) Datasets using the reconstructed humans and objects in *novel human and object poses*. **Bottom Row:** Reconstructed humans animated interacting with *novel objects* in *novel human poses* Note this is a novel application not possible with existing methods.

are not in correspondence to a fixed template. Therefore, we cannot manually define contact Gaussian primitives on the human avatar. Instead, to identify contact Gaussians, we define SMPL contact vertices $V_c^{\text{SMPL}}$ (feet, hips, hands) and assign them to canonical Gaussians by nearest-neighbour search $i^\star = \arg\min_k \|\mathbf{x}_k^C - \mathbf{u}\|_2$ with $\mathbf{u} \in V_c^{\text{SMPL}}$, yielding the index set $\mathcal{I}_c$ - which is a subset of the human Gaussians. Contact frames $t$ are detected from SMPL motion cues (low vertical velocity and downward acceleration). We find displacements $\boldsymbol{\Delta}$ for contact Gaussians. with the objective

$$\mathcal{L} = \lambda_c \sum_{k \in \mathcal{I}_c} \|\mathbf{x}_{k,t}^H - \mathbf{x}_{p(k,t)}^O\|_2^2 + \lambda_p \sum_{k \notin \mathcal{I}_c} h_r(d_\beta(\mathbf{x}_{k,t}^H, \mathcal{G}_O^t))^2 + \lambda_r \|\boldsymbol{\Delta}\|_2^2 + \lambda_t \|\boldsymbol{\Delta}_t - \boldsymbol{\Delta}_{t-1}\|_2^2 \quad (4)$$

where $\mathbf{x}_{p(k,t)}^O$ is the nearest object Gaussian center, $d_\beta(\mathbf{x}) = -\frac{1}{\beta} \log \sum_{p=1}^{N_O} \exp(-\beta\|\mathbf{x} - \mathbf{x}_p^O\|)$ is the soft nearest-neighbour distance to object Gaussians, and $h_r(d) = \max(0, r - d)$ is a hinge margin enforcing separation for non-contact Gaussians. Here we use the sub-index $t$ to denote the timestep.

*Intuitively*, this optimization forces contact Gaussians to snap onto the object surface when contact is expected ($\delta = 1$), pushes non-contact Gaussians at least a margin $r$ away when contact is absent ($\delta = 0$), penalizes large displacements through $\|\boldsymbol{\Delta}\|_2^2$, and encourages temporal smoothness with $\|\boldsymbol{\Delta}_t - \boldsymbol{\Delta}_{t-1}\|_2^2$. The human Gaussian positions are moved as $\mathbf{x}_{k,t}^H + \boldsymbol{\Delta}_{k,t}$ yielding a refined set of Gaussians $\tilde{\mathcal{G}}_H^t$ These are then composited with $\mathcal{G}_O^t = \mathcal{T}(\mathcal{G}_O; \phi_t)$ and jointly rendered, producing photorealistic human–object interactions with coherent contacts.

### 3.7 LEARNING AND NETWORK INITIALIZATION

As all our StyleUnets learn offsets from canonical template, we first train the human StyleUnet for approximately 2000 iterations to map randomly sampled human-position maps to predict zero offsets. We also train the object StyleUnet for a similiar number of iterations to map the pose-independent position map to zero offsets. This ensures that the network predictions start off from zero and all deformation is learnt on top of the Canonical templates. We use the Adam optimizer for all optimization. For learning features (for the canonical Object template) we use two separate MLPs - one maps to geometry parameters - scale, covariance and the other MLP to appearance parameters - color, opacity.

## 4 EXPERIMENTS

In this section we compare our method with recent methods that reconstruct *only* animatable 3D humans from RGB images (Li et al., 2024b; Zhan et al., 2024; Moon et al., 2024; Kocabas et al.,

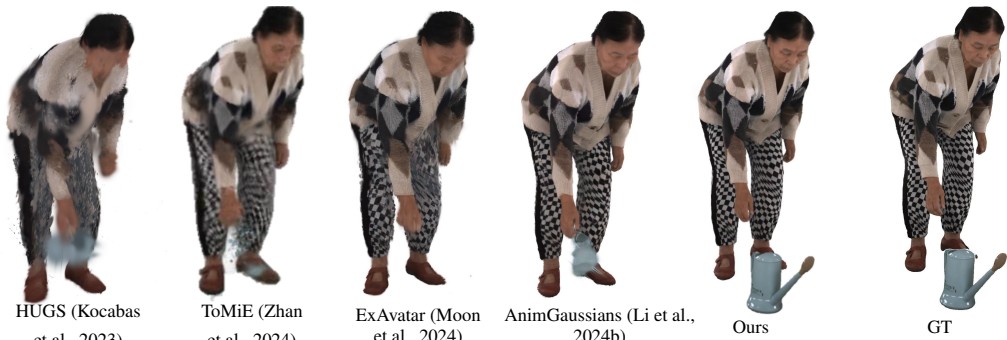

HUGS (Kocabas et al., 2023)    ToMiE (Zhan et al., 2024)    ExAvatar (Moon et al., 2024)    AnimGaussians (Li et al., 2024b)    Ours    GT

Figure 5: Quantiative comparison with unmodified existing methods for animatable human+object reconstruction on Existing methods are unable to reconstruct animatable objects.

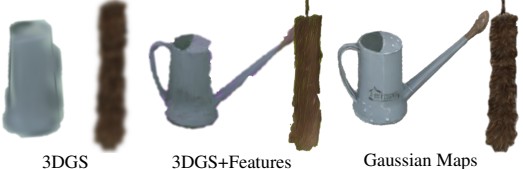

3DGS      3DGS+Features      Gaussian Maps

Figure 6: Gaussian map ablation. Naive 3DGS fails due to occlusion. See Supp Mat for illustration

2023). All existing methods are designed to work only with humans recorded in isolation with *no occlusion* and are *not designed to handle objects*. We conduct two sets of experiments (Sec. 4.1): in the first we use the unmodifed baselines to reconstruct both humans and objects and demonstrate that they fail without significant modification. In the second experiment, we add the object template to the strongest Human reconstruction baseline ((Li et al., 2023b)) without our *Gaussian Map* object formulation. In this experiment, while the human reconstruction quality is comparable to ours, the object is clearly blurry. We then ablate (Sec. 4.3) the various components of our method. In Sec. 4.2, we demonstrate the key novelty of our framework and show how avatars from different datasets can be animated with novel objects to synthesize novel human-object interactions and integrated in 3D scenes to create renderings of avatars interacting with dynamic objects in varied environments.

**Evaluation Dataset:** We use two datasets of human-object interaction: **BEHAVE dataset (Bhatnagar et al., 2022).** It is captured using four Kinect cameras. We use a subset of each sequence for training and test our method on the remaining held-out frames. **DNA-Rendering (Cheng et al., 2023b).** A subset of the DNA-Rendering dataset records people interacting with objects in a studio with 60 cameras. Of the 60 cameras, we use 48 for training and evaluate on the rest.

## 4.1 BASELINES

**Unmodified Baselines:** We compare our approach with ExAvatar (Moon et al., 2024), AnimatableGaussians (Li et al., 2024b), HUGS (Kocabas et al., 2023) and ToMiE (Zhan et al., 2024). As these methods are primarily designed to reconstruct and animate humans recorded in isolation, they need to be modified to deal with objects for a fair comparison. In this experiment, to reconstruct both humans and objects together, we modify the segmentation masks provided in the datasets to include the object along with the humans. All compared methods reconstruct a human using 3DGS, with a canonical space deformed via LBS to posed space and supervised against ground truth pixels. Since

| Subject: | Human | | | Object | | | Full | | |
|---|---|---|---|---|---|---|---|---|---|
| Metric: | PSNR↑ | SSIM↑ | LPIPS↓ | PSNR↑ | SSIM↑ | LPIPS↓ | PSNR↑ | SSIM↑ | LPIPS↓ |
| HUGS (Kocabas et al., 2023) | 22.13 | 0.7674 | 43.71 | 17.54 | 0.6878 | 45.21 | 21.13 | 0.7180 | 42.47 |
| ExAvatar (Moon et al., 2024) | 25.41 | 0.7743 | 37.06 | 16.20 | 0.5752 | 37.91 | 23.18 | 0.7632 | 35.58 |
| ToMiE (Zhan et al., 2024) | 26.85 | 0.7700 | 36.60 | 20.30 | 0.6581 | 35.30 | 25.49 | 0.7790 | 32.33 |
| AnimGau (Li et al., 2024b) | 27.85 | 0.8900 | 32.60 | 23.21 | 0.7550 | 37.40 | 26.49 | 0.8656 | 31.43 |
| AnimGaus+Obj (mod) | 28.16 | 0.9090 | 30.60 | 25.81 | 0.8430 | 31.40 | 27.49 | 0.8956 | 30.43 |
| **Ours** | 28.14 | 0.9174 | 28.88 | 28.63 | 0.8973 | 29.77 | 28.31 | 0.9242 | 29.24 |

Table 1: Quantitative Results on DNA-Rendering evaluated on held-out images. We outperform existing baselines that reconstruct animatable 3D humans — including baselines modified for object reconstruction.

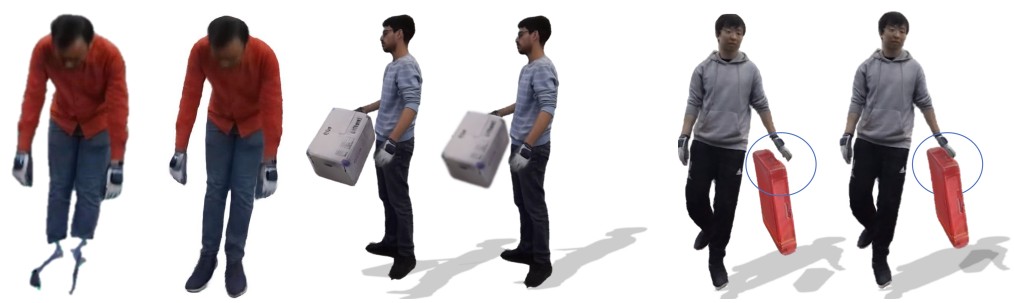

(a) w/o occl., with occl.          (b) i) Ours ii) AnimGaus+Obj          (c) w/o and w contact constraint

Figure 7: Ablation and modified baseline comparison.

| Subject: | Human | | | Object | | | Full | | |
|---|---|---|---|---|---|---|---|---|---|
| Metric: | PSNR↑ | SSIM↑ | LPIPS↓ | PSNR↑ | SSIM↑ | LPIPS↓ | PSNR↑ | SSIM↑ | LPIPS↓ |
| HUGS (Kocabas et al., 2023) | 23.11 | 0.8274 | 33.95 | 19.34 | 0.6378 | 47.43 | 22.12 | 0.7434 | 39.47 |
| ExAvatar (Moon et al., 2024) | 25.41 | 0.8130 | 34.06 | 18.35 | 0.6752 | 38.99 | 22.18 | 0.7820 | 36.58 |
| ToMiE (Zhan et al., 2024) | 23.85 | 0.7840 | 34.60 | 18.51 | 0.6781 | 39.40 | 21.49 | 0.7260 | 37.43 |
| AnimGau (Li et al., 2024b) | 26.85 | 0.8700 | 26.60 | 21.50 | 0.7581 | 34.40 | 24.99 | 0.8160 | 29.43 |
| AnimGaus+Obj (mod) | 27.65 | 0.8710 | 28.70 | 24.50 | 0.8431 | 33.40 | 26.11 | 0.8130 | 29.93 |
| **Ours** | 27.64 | 0.8741 | 28.46 | 26.39 | 0.8732 | 32.47 | 26.92 | 0.8724 | 29.24 |

Table 2: Quantitative Results on the BEHAVE Dataset, evaluated on held-out images not used in training (all human and object poses are unseen in training) We outperform existing baselines that reconstruct animatable 3D humans under occlusion and in presence of objects - including baselines modified for object reconstruction

we force all these methods to reconstruct the object as well, they are forced to modify the canonical Gaussians even to explain pixels detached from the human body. This results in significant artifacts. The object either disappears or is reconstructed as a blob-like surface, as shown in Fig. 5.

We report standard metrics computed on the BEHAVE and DNA-Rendering datasets in Tab. 2 and Tab. 1. To evaluate the reconstruction quality of the human and object, we also report separate standard image metrics for the human and object. Please note that all metrics are computed for *novel poses* on held out *test images unseen during training*. For the DNA-Rendering dataset we report metrics on novel views while for BEHAVE the training and test camera are same as only 4 cameras are available. In Fig. 4 we show results of our method on BEHAVE, DNA-Rendering and NeuralDome.

**Why no comparison with DynamicGaussians (Luiten et al., 2024)?** Note that we focus on animatable humans and objects - while DynamicGaussians and follow ups do reconstruct human-object interaction, these are not controllable. Existing interactions can be replayed from novel views while our method allows for synthesizing humans interacting with novel objects in varied environments.

**Modified Baseline for Objects:** We also conduct a comparison with AnimGaussians (the strongest human reconstruction baseline) where the human and object are reconstructed separately - creating a stronger version of an animatable human baseline. Since we need an object template for object reconstruction, we use the one reconstructed and tracked using the feature representation, without the *pose-independent* Gaussian Map formulation from our method. In this experiment, while the human reconstruction quality is comparable to ours, without our Gausian Map formulation, the object is clearly blurry (Fig.7b). We report PSNR for this experiment on BEHAVE and DNA for novel-pose synthesis in Tab 2 and 1 under the row **AnimGaus+Obj (mod)**.

| Subject: | DNA-01 | | | DNA-02 | | | BEH-01 | | |
|---|---|---|---|---|---|---|---|---|---|
| Metric: | PSNR↑ | SSIM↑ | LPIPS↓ | PSNR↑ | SSIM↑ | LPIPS↓ | PSNR↑ | SSIM↑ | LPIPS↓ |
| w/o Features | 26.11 | 0.9674 | 40.95 | 27.54 | 0.9678 | 46.43 | 25.00 | 0.9218 | 59.47 |
| w/o Obj Map | 30.41 | 0.9743 | 24.06 | 33.20 | 0.9752 | 28.99 | 26.18 | 0.9232 | 35.58 |
| w/o Occ Loss | 29.12 | 0.9727 | 26.58 | 32.94 | 0.9695 | 36.04 | 26.63 | 0.9141 | 41.76 |
| w/o Contact | 32.34 | 0.9664 | 20.91 | 32.63 | 0.9673 | 25.87 | 27.44 | 0.9674 | 28.86 |
| **Full** | 32.64 | 0.9774 | 20.88 | 33.63 | 0.9773 | 25.77 | 27.64 | 0.9574 | 28.46 |

Table 3: Ablation Study regarding the various components of our method on the DNA-Rendering and BE-HAVE datset. As the table indicates, each component in our method adds to the performance and our method cumulatively yields best results. Each line the table corresponds to an experiment detailed in Sec. 4.3

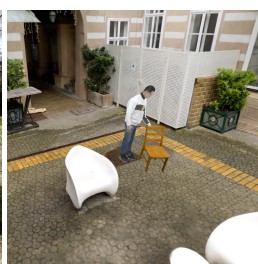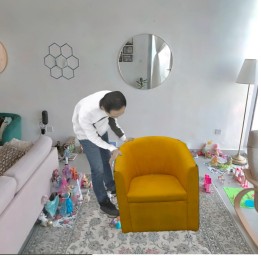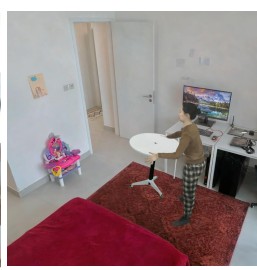

Figure 8: Our method allows for combination of dynamic object movements with avatars reconstructed without objects (here from AvatarX) and placed in 3D scenes.

### 4.2 ANIMATING HUMANS WITH NOVEL OBJECTS AND COMPOSITION WITH SCENES

Our framework allows reanimating a reconstructed human identity reconstructed from one set of videos with object Gaussians, human poses and object poses of another sequence. This allows, *for the first time* in the context of photo-real neural rendering, retargeting of human object interaction from one sequence to another. We find it necessary to optimize for contacts to ensure human-object interaction remains plausible (see Sec. 3.6) In Fig. 4, we qualitatively demonstrate this ability of our method by animating a reconstructed avatar to interact with novel objects. In Fig. 8, we demonstrate that these dyanmic interactions can be placed in diverse 3D environments and thus for the first time creating photoreal virtual Avatars that interact with dynamic objects in their environments.

### 4.3 ABLATION STUDIES

In Tab. 3, we evaluate our design choices. Each method component improves the results. **Contact Constraint:** Here we evaluate contact constraint utility on same human+object using GT images but for novel poses. See **w/o Contact** in Tab. 3. The improvement for same human+object is minimal but pronounced for novel human-object interaction. See Fig. 7c. **Obj Gaussian Map:** In this setting, we do not refine the final object reconstructions using Gaussian maps but generate final renderings using the coarse object template. As shown in Tab. 3 (**w/o Obj Map**) and Fig. 6, the Gaussian Map based Gaussian parameter prediction allows for the reconstruction of high-frequency details. **Features:** Here we optimize the locations of the 3DGS parameters directly. Naive 3DGS fails due to significant occlusion patterns in the captured videos. See **w/o Features** experiment in Tab. 3. Without features the template reconstruction does not converge. Please see Supp Mat for illustration. **Occlusion Loss:** We switch off the occlusion-aware loss. This forces the occluded regions of the human and object in the rendered image to be part of the background, which degrades reconstruction quality as shown in Fig. 7a and the **w/o Occ Loss** experiment reported in Tab. 3.

## 5 CONCLUSION

We have presented GASPACHO, a method to reconstruct photorealistic human and object interactions from multi-view 2D images, which is capable of human, object and camera pose control. Different from prior art, which only generates photo-real *human actions in empty space* our approach completely decouples the object from the human, enables independent object and human pose control and *for the first time allows for the synthesis of photo-real human animations interacting with novel, previously unseen objects* and *placement of such interactions in diverse 3D scenes* and thus for the first time creating photoreal virtual Avatars that interact with dynamic objects in their environments. We evaluate GASPACHO quantitative and qualitatively on BEHAVE and DNA-Rendering datasets. Our method does make assumptions about the nature of the 3D scenes: i.e we only model one dynamic object and assume that its motion can be explained using only a rigid transform. Future extensions include the reconstruction of humans and objects not only in a lab setting but also from monocular RGB videos collected in the wild. We also hope to extend our work to build autonomous agents that interact not only with small objects but also with large 3D scenes.

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
