# OpenReview forum: "GASPACHO: Gaussian Splatting for Controllable Humans and Objects"
_ICLR.cc/2026/Conference — Submitted to ICLR 2026_

### Official Review · Reviewer_ttja · 2025-10-24

**Soundness:** 2
**Presentation:** 1
**Contribution:** 2
**Rating:** 4
**Confidence:** 3

**Summary:**

This work focuses on modeling human-object interactions from multi-view RGB video with Gaussian Splatting. While previous work often under neglect the dynamic movement of object, this work introduce a coarse-to-fine pipeline for reconstructing dynamic objects. The moded humans and objects can be animated to synthesize novel interactions. It also introduce human-object contact constraints in Gaussian splace to ensure proper concats when 3D Gaussian humans are animated to interact with objects.

**Strengths:**

1. While the baseline methods cannot accurately reconstruct objects, the proposed method demonstrates much better visual performance than baselines.

2. The introduced components are vaild through ablation study.

3. Methods are introduced with mathematical equations.

**Weaknesses:**

1. The process of obtaining position maps from input images should also be included in Figure 2 for completeness.

2. The application of reconstructing human interact with novel objects seems fancy, but it can be achieved with Gaussian editing or segmention baselines. The advancement of using the proposed method is not mentioned.

3. The introduction of baselines in ablation studies is confusing. Meanwhile the paper uses a lot of bold font without a clear pattern. The format of this paper need to be further polished.

4. Figure 3 seems to be redundant, given the information are included in figure 2.

Missing related work:
[1] Wang, Xiaoyuan, et al. "HoliGS: Holistic Gaussian Splatting for Embodied View Synthesis." arXiv preprint arXiv:2506.19291 (2025).

Minor weakness: Some texts are overlap with figures. Meanwhile some labels of figure images are introduced in the caption, which could be clearer if put them directly under the images, like Figure 7. Meanwhile the space between text looks weird. Hope this can be adjusted in the revision.

**Questions:**

1. Is this method able to model multple humans and objects in one scene?

2. Why traditional 3DGS fails under natrually occurring human-object occlusions, but feature-based planes can manage that? Are other baselines also feature-based? If not, the contribution of feature-based representation seems to be more important.

---

> ### Author Response · Authors · 2025-11-19
>
> We thank the reviewer for their feedback. We have tried to incorporate it all into the revised manuscript.
>
> **1) Missing reference**: we thank the reviewer for pointing us to the missing reference. We have included it in the revised manuscript (l114)
>
> **2) Text overlap and labels**: We appreciate the feedback. We have updated figure 2 and figure 5 in the paper to remove any text overlap and we have also included labels in Figure 6 to make it clear what each figure refers to - as the reviewer suggested.
>
> **3) Weird Space**: we have rectified this as well - especially in the experiments section. We thank the reviewer for the feedback. If the reviewer has further feedback for us regarding formatting we would be happy to incorporate the reviewer’s suggestion.
>
> **4) Bold font** : We have removed all bold font from the text in the experiments section except to indicate the start of a new paragraph or to highlight a Dataset or experiment. We appreciate the feedback. If the reviewer wishes for us to iterate further on the manuscript we would be happy to do so.
>
> **5) Obtaining position maps from input images and redundancy of figure 3:** We thank the reviewer for this suggestion. Here we want to highlight that figure 3 and figure 2 are complimentary - figure 2 does not explain how position maps are computed - as we thought it would make the figure overcrowded. As such we have split the figure - and figure 3 includes the essential information about getting position maps. To obtain position maps, as figure 3 tries to highlight - we use two fixed cameras which project the canonical template gaussians to obtain 2D projections of the object template. This projection - (as figure 3 tries to explain) provides a mapping between 2D maps and 3D object Gaussian locations. Furthermore we have updated the caption of Figure 3 to also highlight that this figure explains how position-maps are obtained. Each pixel in the object map by storing the position of its corresponding object Gaussian yields a position map. We hope this clarifies the purpose of the 2 figures. If the reviewer has further feedback regarding the figures we would be happy to incorporate it.
>
> **6) Application of humans interacting with novel objects (and existing Gaussian segmentation and editing baselines)**: Here we want to highlight that we are proposing the first method where the human-object interaction can be explicitly controlled to generate photoreal renderings of humans interacting with objects (optionally composited with 3D scenes). To the best of our knowledge our method goes well beyond what is possible with existing Gaussian segmentation or editing baselines - to the best of our knowledge no method exists which allows for generating controllable photoreal (in the context of neural rendering) renderings of humans moving chairs, lifting objects composited with varied 3D scenes. We invite the reviewer to watch the supp video from 0:40 to 1:40 - where we demonstrate for the first time how Gaussian Splatting (or neural rendering in general) can be used for controllable generation of human-object interactions composited with 3D scenes. If the reviewer wishes for us to compare with a specific existing baseline which allows for such results, we would be happy to do so.
>
> **7) Explanation regarding ablation studies** : In the beginning of the experiments section we have tried to explain how we adapt existing baselines which only work for controllable human reconstruction to the task of reconstructing controllable 3DGS models of humans and objects. In the section on unmodified baselines, we use existing baselines for controllable human reconstruction for joint human+object reconstruction by simply expanding the masked regions of the images (from just human to human+object) without modifying the underlying algorithm for 3DGS reconstruction. As this forces the same Gaussians to explain human and object pixels these baselines fail. In the second experiment on “modified baselines” we modify the strongest human-only multiview baseline - Animatable Gaussians - with an object template as well - i.e we split the object and human Gaussians but we do not use our idea of Gaussian maps for object reconstruction. As we demonstrate in Table 1 and 2 and Figure 7, the reconstructed object does not match the quality of our reconstruction. We further want to highlight that our method goes beyond existing baselines by also allowing for the generation of controllable renderings of humans interacting with novel objects - composited with 3D scenes
>
> **8) Multiple humans**: We want to highlight that we explicitly acknowledge in the conclusion that our method only works with one human and one object. If the reviewer wishes to make it more explicit and clear in the text we would be happy to incorporate this suggestion as the reviewer wishes

---

> ### Author Response · Authors · 2025-11-19
>
> **9) Why 3DGS fails and features help**:  We have tried to explain in the text that naive 3DGS does not contain any smoothness priors or any regularizers . We have included a longer explanation in the supplementary. We paste the text here “The original 3DGS algorithm is designed to work with images of complete objects in static scenes. We observe that 3dGS doesn’t converge when used to reconstruct objects under occlusion using object-only pixels. This motivates using features. They act as a regularizer and a smoothness prior for reconstruction under occlusion. This observation has been noted in prior work as well Mihajlovic et al. (2024); Kocabas et al. (2023) which use features as a smoothness prior for 3DGS reconstruction. In our problem, even under dense camera setups, the object is occluded from a significant number of cameras. Fig.1 shows observed pixels of one object in some images from different cameras at a single timestep.” If the reviewer thinks it’s more important to include this text in the main paper instead of the supplementary we would try our best to include it in the main manuscript.
>
> We once again thank the reviewer for their feedback - which has greatly improved the manuscript. We hope we have addressed all the concerns raised by the reviewer. If there is anything else we can provide - experiments, clarifications, revisions - for the reviewer to increase their rating, we would be happy to provide this.

---

> > ### Comment · Reviewer_ttja · 2025-11-27
> >
> > Thanks authors for the rebuttal. Now I learn the reason for using feature-based 3DGS. I am still wondering why the proposed method can only be used for one human and one object, from a question perspective, not a weakness. The manipulation of objects can be done with 3D Gaussian segmentation methods, including Gaussian Grouping [1], SAGA [2], OmniSeg3D [3], Click-Gaussian [4], or language-embedding methods, including LERF [5], LEGaussians [6], LangSplat [7], modify its position and orientation, and combine with human reconstruction pipelines. So even if this paper is the first to achieve human-object interaction, I still think it is more engineering than research. Meanwhile, in the video, I do see artifacts like the human's hand and the table are not in contact (1'29), and the human's hand passes through the object (1'23, 3'32). I know that for a Gaussian-based task, these artifacts may be inevitable, but that still shows the current performance is far from photo-realistic rendering. So I would maintain my score for now.
> >
> > I still want to point out that the paper's format needs further polishing. The titles in Related Works, Sec. 3.3 are not consistent. The word "Intuitively" in the Method section is bold for no obvious reason. Figure 4 Top Row, Bottom row., etc.
> >
> > [1] Ye, Mingqiao, et al. "Gaussian grouping: Segment and edit anything in 3d scenes." European conference on computer vision. Cham: Springer Nature Switzerland, 2024.
> >
> > [2] Cen, Jiazhong, et al. "Segment any 3d gaussians." Proceedings of the AAAI Conference on Artificial Intelligence. Vol. 39. No. 2. 2025.
> >
> > [3] Ying, Haiyang, et al. "Omniseg3d: Omniversal 3d segmentation via hierarchical contrastive learning." Proceedings of the IEEE/CVF Conference on Computer Vision and Pattern Recognition. 2024.
> >
> > [4] Choi, Seokhun, et al. "Click-gaussian: Interactive segmentation to any 3d gaussians." European Conference on Computer Vision. Cham: Springer Nature Switzerland, 2024.
> >
> > [5] Kerr, Justin, et al. "Lerf: Language embedded radiance fields." Proceedings of the IEEE/CVF international conference on computer vision. 2023.
> >
> > [6] Shi, Jin-Chuan, et al. "Language embedded 3d gaussians for open-vocabulary scene understanding." Proceedings of the IEEE/CVF Conference on Computer Vision and Pattern Recognition. 2024.
> >
> > [7] Qin, Minghan, et al. "Langsplat: 3d language gaussian splatting." Proceedings of the IEEE/CVF Conference on Computer Vision and Pattern Recognition. 2024.

---

> ### Author Response · Authors · 2025-11-27
>
> We appreciate the detailed answer from Reviewer ttja. The statement that our work "is more engineering than research" is made because "the manipulation of objects can be done with 3D Gaussian segmentation methods [...] or language-embedding methods". We respectfully disagree and would like to point out several reasons:
>
> - Listed references [1-7] address how to separate static scenes into 3D object instances with masks consistent across views from open-world segmentation or text prompts, enabling the creation of a Gaussian model for a given object. This is not related at all to the problem we tackle and not sufficient for a methodology that can reconstruct dynamic human-object interactions - in fact these papers have *no relation at all to 3D humans*. The goals on our set-up are a) to reconstruct animatable human+object templates from multiview input videos, b) to controllably animate these reconstructed templates to generate novel human+object interactions (optionally composited with a new 3DGS scene). As Tab 1 and Tab 2 of the manuscript show (and acknowledged by three other reviewers), our method leads to SoTA performance for the task of joint controllable human+object reconstruction on multiple datasets. In the human-object interaction problem set-up, the scene is *dynamic* and the object is most of the time *heavily occluded*, such that classical naive reconstruction of objects fail (see Fig 6 in the paper). Our novel object reconstruction pipeline addresses this problem with innovative technical solutions: features space template, 3D tracking of objects, and pose-independent Gaussian maps. Proposed references do not address this.
>
> - The claim that "the application of reconstructing human interact with novel objects seems fancy, but it can be achieved with Gaussian editing or segmentation baselines." made by the reviewer is not supported by any evidence from previous literature. The applications shown in these listed papers are significantly different from Human Object Interaction and integrating them in a HOI pipeline is very far from being straightforward, if feasible at all. Can the reviewer please back up these claims with factual evidence ?
>
> - The boundary between engineering and research in our field is not a clear one that has solid consensus, and is largely a subjective assessment that is not relevant to ICLR reviewing guidelines (e.g. "4 key questions"). Our paper is closer to application than other ICLR papers, which is not ground for a negative evaluation. We present original research which includes several novel technical contributions thoroughly validated experimentally, and we would like our work to be evaluated on these grounds.
>
> We thank the reviewer for feedback regarding formatting. We have updated the manuscript. The caption of Figure 4 still uses top row and bottom row because we think putting this text in the figure makes the overall figure look significantly overcrowded hence we deliberately put the text in the caption here below figure.

---

### Official Review · Reviewer_H4ie · 2025-10-31

**Soundness:** 2
**Presentation:** 2
**Contribution:** 2
**Rating:** 6
**Confidence:** 3

**Summary:**

This paper presents GASPACHO, a method for generating photorealistic and controllable renderings of human–object interactions from multi-view RGB video. The approach models both the human and the interacting object as distinct sets of Gaussian primitives. It introduces a novel formulation for objects, learning Gaussians on an underlying 2D surface manifold to capture fine-grained details. The method also proposes a contact constraint in Gaussian space to regularize human-object relations and enable physically plausible animations. Experiments on the BEHAVE, NeuralDome, and DNA-Rendering datasets demonstrate high-quality reconstructions under occlusion and the ability to synthesize novel, controllable interactions.

**Strengths:**

- The paper simultaneously reconstructs humans and interacting objects under occlusion, advancing beyond human-only Gaussian methods.
- Introducing contact constraints in Gaussian space improves physical plausibility and reduces interpenetration during animation.
- The framework enables photorealistic and controllable synthesis of human–object interactions across diverse scenes and viewpoints.

**Weaknesses:**

- The quantitative results presented in Table 1 and Table 2  are difficult to parse quickly. The best-performing result in each column is not bolded or otherwise highlighted, forcing the reader to manually scan all numbers to identify the state-of-the-art.
- Although the paper introduces a contact constraint in Gaussian space, the contact quality remains suboptimal. Hands and other body parts often penetrate the object surfaces, suggesting that the constraint is weak or insufficiently enforced during animation.
- The method makes a strong assumption that it only models "one dynamic object" and that this object's motion "can be explained using only a rigid transform". This is a significant limitation, as it excludes a vast category of common human-object interactions involving non-rigid or articulated objects, such as interacting with blankets, clothing, ropes, or laptops.

**Questions:**

- In Figure 2, what does the gray translucent mask represent?
- The contact constraint (Sec 3.5) is defined by mapping a fixed set of SMPL vertices ("feet, hips, hands")  to the nearest Gaussians. How does the method handle or enforce plausible contact for other body parts not in this list?

---

> ### Author Response · Authors · 2025-11-19
>
> We thank the reviewer for the feedback. We are encouraged to see that they recognize that we address a novel problem which enables a new application not previously possible. We have tried to address the issues raised by the reviewer below.
>
> **1) Tables difficult to parse** We have updated the manuscript to make the tables easier to parse - with color based highlighting. We thank the reviewer for this feedback.
>
> **2) Gray mask**: The gray mask represents regions which are outside the 2D map to 3D Gaussian mapping - i.e there is no Gaussians attached at these pixels - whatever the neural network predicts for these grey regions, these pixels are masked out (this is possible as the mask for the canonical 2D map for both objects and humans is known a priori) and we do not lift these pixels to 3D Gaussians. We have updated the manuscript and the caption of the figure to make it explicit. We thank the reviewer for this feedback.
>
> **3) Limitations**: We acknowledge that our method has limitations - indeed we have mentioned them ourselves in the conclusion section. The fact that there are many exciting open challenges ahead for human-object 3d vision research, e.g. articulated objects, multiobject processing, in-the-wild capture; should not subtract from the fact that we have presented a method that makes solid progress towards those ambitious goals. We want to highlight that our work goes beyond prior art, which restricts itself to only modelling controllable humans. As we demonstrate in Table 2, and Table 1 our method allows for much better joint  human+object reconstruction on three datasets. And as we show in the supp video from 0:40 to 1:40 our method for the first time allows for controllable interaction synthesis of 3DGS humans interacting with various objects composited with 3DGS scenes. We argue that the results showcased in our paper are new, appealing and of value to the wider community to further solve remaining challenges of human-object animation.
>
> **4) Contacts and fixed set of regions**: We acknowledge that our method restricts itself to modelling contacts between human hands/hips/feet with 3D objects - this does not allow our method to model motions such as a ball being hit against the human belly- but we want to highlight that for the vast majority of human-object interactions, the hands, feet and the hips are the most salient parts. And we also want to highlight that human-object contact modelling is an active area of research. We, to the best of our knowledge, present the first method which uses 3DGS to generate controllable renderings of humans interacting with varied objects composited with 3D scenes. We again invite the reviewer to watch the supplementary video from 0:40 to 1:40 - where we demonstrate for the first time how Gaussian Splatting (or neural rendering in general) can be used for controllable generation of human-object interactions composited with 3D scenes. We hope that our ideas serve as a catalyst for further research in this direction.
>
>
> We once again thank the reviewer for their feedback. We hope we have addressed all the concerns raised by the reviewer. If there is anything else we can provide - experiments, clarifications, revisions - for the reviewer to increase their rating, we would be happy to provide this.

---

### Official Review · Reviewer_zCUP · 2025-11-01

**Soundness:** 3
**Presentation:** 3
**Contribution:** 2
**Rating:** 6
**Confidence:** 3

**Summary:**

This paper proposes GASPACHO, a 3D Gaussian-based neural rendering framework that jointly reconstructs animatable humans and dynamic objects from multi-view RGB sequences, with the explicit goal of enabling controllable human–object interactions under novel human/object poses and viewpoints. The core ideas are: (i) learn pose-dependent Gaussian maps for humans and pose-independent Gaussian maps for objects, anchored to canonical templates; (ii) use a composition- and occlusion-aware loss during training so that occluded human regions are not penalized improperly; and (iii) introduce a Gaussian-space contact refinement that adjusts a sparse set of “contact Gaussians” to promote physically plausible human–object contact at test time. Compared to prior 3DGS avatar work that typically reconstructs humans in isolation, GASPACHO explicitly separates and animates both entities and demonstrates novel cross-sequence retargeting.

**Strengths:**

1. Addresses controllable human–object interactions instead of human-only avatars, enabling cross-sequence retargeting and scene composition.
2. Introduces pose-independent Gaussian maps for rigid objects that stabilize pose optimization and sharpen textures; principled occlusion-aware losses mitigate erroneous supervision; contact refinement improves plausibility.
3. Demonstrates consistent quantitative gains over strong baselines and compelling qualitative compositions across multiple datasets.

**Weaknesses:**

1. Restricts to one rigid object, and there is no support for non-rigid objects or multiple objects, limiting applicability to richer HOI scenes.
2. Requires SMPL poses and object masks; sensitivity to pose/mask errors is not thoroughly analyzed. A robustness study would be informative.
3. While the paper explains why some dynamic 3DGS systems are not controllable, additional qualitative side-by-side or an explicit metric for controllability would strengthen the case. Also, report compute/time/memory for training/inference to contextualize practicality.

**Questions:**

1. What are training and inference times, memory footprints, and Gaussian counts for typical sequences (by dataset)?
2. How does performance degrade with noisy SMPL, imperfect masks, or fewer cameras? Can the method recover without accurate masks?

---

> ### Author Response · Authors · 2025-11-19
>
> We thank the reviewer for their feedback. We are encouraged to that they recognize we address a novel problem and that our method generates improvements over existing baselines. Below we try to address the concerns raised by the reviewer.
>
> **1) Masks and SMPL**: All our experiments are conducted on datasets collected in indoor multiview capture setups - BEHAVE, DNA-Rendering, NeuralDome. In all these datasets, we find that SoTA segmentation methods - especially Sam2 - are very accurate. We also want to highlight we don’t have real ground-truth segmentation masks anywhere - we are using segmentation masks from these trained networks. Under these settings with the predicted segmentation masks we don’t see significant degradation in performance. But in theory if the masks are significantly incorrect, there would of course be degradation in reconstruction quality as the same 3D gaussians would be forced to explain regions in the background or the foreground. This is an interesting problem for in-the-wild 3D reconstruction but our setting is different - i.e controllable reconstruction and animation of humans and objects - hence we do not study this problem. To address the reviewer’s question we tried out an experiment where we dilated the masks for both the human and the object using a 10x10 kernel. We then run our method using these dilated masks. For a BEHAVE sequence this causes PSNR to drop from 25.6 to 21.2 for novel-pose evaluation. If the reviewer has another experiment in mind regarding mask analysis we would be happy to provide this.
> We also want  to highlight that we are using SMPL poses estimated from RGB images for learning - i.e we don’t use mocap derived SMPL. In the setting we are working with -  multi-view capture in indoor environment - multiview SMPL pose estimates from RGB are very accurate, hence this is another problem we did not focus on. We acknowledge that this is an interesting problem for in-the-wild 3D reconstruction but our setting is different. We also want to highlight that we have used the same SMPL estimates and masks for evaluating our method and our baselines, so whatever improvement/degradation happens in our performance due to changes in masks or SMPL parameters, would also be reflected in our baselines.
>
>
> **2) Fewer cameras**: We want to highlight that we report results for DNA-rendering and BEHAVE datasets. BEHAVE only provides 4 cameras (a very sparse multiview setup). While PSNR in general for BEHAVE is lower than PSNR for DNA-Rendering - as Table 1 and Table 2 show our method outperforms all baselines for human+object reconstruction. We further want to highlight results visualized in Figure 4 - and in the supplementary for BEHAVE. Though they are slightly worse than those obtained for DNA-Rendering (48 cameras) they are reasonably photoreal and again we want to highlight better than existing baselines. Thus we are confident in claiming that for very sparse multi-view setups our method performs reasonably well. We also provide a new experiment for a DNA-Rendering sequence where we run our method on 48 24 and 12 cameras. Total human+object PSNR drops from 28.1 (48 cameras) to 27.9 (24 cameras) to 27.3 (12 cameras) -- thus demonstrating that our method is robust to sparsity of cameras.
>
> | Cameras | 48 | 24 | 12 |
> |---------|----:|----:|----:|
> | PSNR    | 28.1 | 27.9 | 27.3 |
>
>
>
> **3) Controllability**: We present a side by side comparison with our method compared to a non-controllable dynamic 4dGS method [DynamicGaussians] in Figure 5 of the revised supplementary manuscript. We are unaware of any metric that would allow us to quantify controllability explicitly, as methods that are not controllable generally loose alignment with held out testing set with novel (human or object) poses and thus the performance on pixel fidelity metrics e.g. PSNR would mostly be dominated by this. We remain open to additional suggestions in terms of metrics to further illustrate this difference, however we note that it is fairly uncommon in literature to compare methods that are controllable to purely dynamic (non-controllable) ones as the problem set-up and complexity is fundamentally different, and the evaluation protocol for each of them requires different testing sets (i.e. same sequence with novel camera views, vs different sequences with both same and novel camera views).

---

> ### Author Response · Authors · 2025-11-19
>
> **4) Training time/memory**:  As we train two separate networks for the object and human, our training takes longer to converge than a human-only baseline. Training a human only baseline (AnimtableGaussians) takes about 72 hours for convergence; while our full method human+object takes approximately 84 hours - the additional 12 hours are for learning the object StyleUnet. The object network converges much faster as it is only modelling rigid deformations and sharp textures - unlike the pose-dependent human net which learns pose-dependent deformations. The contact based optimization takes approximately 180 seconds for a motion sequence of 30 seconds; but this is per instance - after training. We hope this addresses the question. The networks take about 10GB of VRAM on NVIDIA A100/V100 GPUs
>
> **5) Gaussian count**: For humans we typically use 300k Gaussians (across datasets) and for objects Gaussians between 30k to 80k. Here we want to highlight that our aim is not efficiency but to show that 3DGS can be used (for the first time) for controllable generation of humans  interacting with objects.
>
> **6) Limitations**: We acknowledge that our method has limitations - indeed we have mentioned them ourselves in the conclusion section. The fact that there are many exciting open challenges ahead for human-object 3d vision research, e.g. articulated objects, multiobject processing, in-the-wild capture; should not subtract from the fact that we have presented a method that makes solid progress towards those ambitious goals. We want to highlight that our work goes beyond prior art, which restricts itself to only modelling controllable humans. As we demonstrate in Table 2, and Table 3 method allows for much better joint  human+object reconstruction. And as we show in the supp video from 0:40 to 1:40 our method for the first time allows for controllable interaction synthesis of 3DGS humans interacting with various objects composited with 3DGS scenes. We argue that the results showcased in our paper are new, appealing and of value to the wider community to further solve remaining challenges of human-object animation.
>
>
> We once again thank the reviewer for their feedback. We hope we have addressed all the concerns raised by the reviewer. If there is anything else we can provide - experiments, clarifications, revisions - for the reviewer to increase their rating, we would be happy to provide this.

---

### Official Review · Reviewer_Rw2b · 2025-11-01

**Soundness:** 3
**Presentation:** 1
**Contribution:** 2
**Rating:** 4
**Confidence:** 3

**Summary:**

This paper proposes GASPACHO, a framework that reconstructs animatable humans and objects from multi-view posed videos. It reconstructs humans and objects separately using distinct sets of 3D Gaussians, enabling controllable rendering of novel human–object interactions under different poses and from novel camera viewpoints. For object reconstruction, it employs pose-independent Object Gaussian Maps for efficient reconstruction. Additionally, it introduces a contact constraint in Gaussian space to regularize human–object relationships. Experimental results demonstrate that the proposed method achieves SOTA performance and produces reasonable novel view renderings.

**Strengths:**

- The proposed work addresses the problem of reconstructing both humans and objects from multi-view posed videos and rendering novel human–object interactions with new poses, which is interesting.

- GASPACHO reconstructs objects using pose-independent object maps, enabling efficient and robust object reconstruction.

- The proposed method achieves comparable performance to previous SOTA methods on human reconstruction and animation tasks.

**Weaknesses:**

- The writing quality is poor, making the paper difficult to read and understand. Moreover, many technical details are unclear:
    - Numerous mathematical notations are introduced without proper explanation, making it difficult to interpret the method section.
    - In Line 265, the term “ground-truth images” is confusing, as ground truth usually refers to evaluation images; input images would be more appropriate.
    - In Line 199, since the Gaussians are learned later, how are their locations projected onto the planes?
     - In Line 202, it is unclear how the frame with minimal occlusion is determined.
    - The process of obtaining the object template is poorly described; providing pseudocode for this step would improve clarity.
    - The initialization and training procedure of the StyleUNet network for producing Gaussians are not explained.

- The paper includes only one quantitative comparison with previous methods, which is insufficient to validate the results.

- The paper lacks a discussion of the proposed method’s limitations.

**Questions:**

- The main issue lies in the clarity of the paper’s writing and presentation.

- How do the training and inference times compare with the baselines?

---

> ### Author Response · Authors · 2025-11-19
>
> We thank the reviewer for their feedback. We are extremely encouraged to see they recognize that our method addresses a novel problem and introduces a novel idea about 3D human and object reconstruction and that we present good visual results. We also appreciate the reviewers feedback regarding writing. We have edited the manuscript with improved writing and clarity, following reviewer recommendations.
>
> **1) Writing**: We thank the reviewer for the feedback. We have significantly rewritten some sections of our method - especially 3.1, 3.2, 3.3 and 3.4 to make the explanation more intuitive - reliant on text and figures - without involving dense mathematical notations. We hope this aids understanding. We have also tried to simplify mathematics used throughout the paper. We very much hope this addresses the concerns about writing the reviewer raised.
>
> **2) Target vs Ground-truth**: We thank the reviewer for this feedback. We have updated the writing and replaced the word ground-truth images with “target” or “input” images in the manuscript, as the reviewer suggests .
>
> **3) Minimal occlusion frame**: We have updated the manuscript (Sec 3.1) (l212) to include further description of the minimal occlusion frame - we compute this using object segmentation masks and by finding the temporal frame with the largest number of object pixels. We hope this addresses the question; if there are still some queries we would be happy to address them.
>
> **4) Locations projected to features and object template**: We thank the reviewer for this feedback. We have updated the manuscript (the whole section 3.1) to include further details about the template reconstruction and feature projection. We have also included pseudo-code in the supplementary (Algorithm 1) and a new figure (Fig. 3) in the supplementary. The general idea for object template reconstruction is that we optimize the Gaussian parameters which match the segmented object pixels in one set of multiview images (i.e at one temporal frame). This setup closely matches standard 3DGS reconstruction but with features and segmented object images as targets - i.e we do learn the Gaussian parameters - opacity, color, covariance and scale but parameterized by Gaussian positions and the features. We query a learnt feature representation at these optimized positions and map them to Gaussian properties. This yields a coarse template. We invite the reviewer to look at Fig 3 provided in supplementary for further understanding of the feature based representation. For feature projection we use standard ideas about orthographic projection and bilinear interpolation - also used previously in Kockabas et al. (HUGS) and Mihajlovic et al (SplatFields) - our contribution is to show that such a representation provides better results for reconstruction of objects under occlusion. We thank the reviewer for this feedback; we hope the updated manuscript better explains the part of the algorithm.
>
> **5) Initialization and training of StyleUnet**: We again thank the reviewer for this feedback. We have updated the manuscript to include a new section 3.7 on network training and about StyleUnet initialization. We hope this addresses the concerns raised by the reviewer.

---

> ### Author Response · Authors · 2025-11-19
>
> **6) Quantitative comparisons (at least five not one)** Here we want to highlight that in the manuscript, we provide quantitative evaluations on two datasets in Table 1 (DNA-Rendering) and 2 (Behave), each of them compared with 5 other SoTA methods (and not just one as the reviewer indicates), across three metrics, and evaluated across three different image regions (human, object and full). We also want to highlight that we have provided figures of comparisons with multiple methods in the main paper. We also want to point the reviewer towards Table 1 in the supplementary for numbers on the NeuralDome dataset. We note that we require datasets that contain multiview coverage of humans and objects, and to the best of our knowledge we are not aware of any other dataset publicly available that can support our experimental set-up. We have also tried our level best to improve the SoTA multiview controllable human reconstruction baseline by including an object template in the reconstruction (using parts of our contribution); and again we want to highlight that our full reconstruction results are better than the ones obtained by all existing baselines. We are keen to improve our work, and remain open to include any specific suggestion that the reviewer might be able to provide in the remainder of the discussion phase.
>
> Here we provide the numbers from Table 2 for the reviewer to scan easily
> | Method        | Human PSNR ↑ | Human SSIM ↑ | Human LPIPS ↓ | Object PSNR ↑ | Object SSIM ↑ | Object LPIPS ↓ | Full PSNR ↑ | Full SSIM ↑ | Full LPIPS ↓ |
> |---------------|--------------:|-------------:|--------------:|--------------:|--------------:|---------------:|------------:|------------:|-------------:|
> | HUGS          | 23.11 | 0.8274 | 33.95 | 19.34 | 0.6378 | 47.43 | 22.12 | 0.7434 | 39.47 |
> | ExAvatar      | 25.41 | 0.8130 | 34.06 | 18.35 | 0.6752 | 38.99 | 22.18 | 0.7820 | 36.58 |
> | ToMiE         | 23.85 | 0.7840 | 34.60 | 18.51 | 0.6781 | 39.40 | 21.49 | 0.7260 | 37.43 |
> | AnimGaus      | 26.85 | 0.8700 | **26.60** | 21.50 | 0.7581 | 34.40 | 24.99 | _0.8160_ | _29.43_ |
> | AnimGaus+Obj  | **27.65** | _0.8710_ | 28.70 | _24.50_ | _0.8431_ | _33.40_ | _26.11_ | 0.8130 | 29.93 |
> | **Ours**      | _27.64_ | **0.8741** | _28.46_ | **26.39** | **0.8732** | **32.47** | **26.92** | **0.8724** | **29.24** |
>
>
>
> **7) Training time / Baseline** : As we train two separate networks for the object and human, our algorithm takes longer to converge than a human-only baseline. Training a human only baseline (AnimtableGaussians) takes about 72 hours till convergence; while our full method human+object takes approximately 84 hours - the additional 12 hours are for learning the object StyleUnet. The object network converges much faster as it is only modelling rigid deformations and sharp textures - unlike the pose-dependent human net which learns pose-dependent deformations. The contact based optimization takes approximately 180 seconds for a motion sequence of 30 seconds; but this is per instance - after training. We hope this addresses the question.
>
> **8) Discussion of limitations (provided in conclusion)** : Here we want to highlight that we list several limitations of the method in the conclusion - we explicitly mention that we model one dynamic object and assume that its motion can be explained using only a rigid transform. If the reviewer wishes for us to expand upon this - such as mentioning elastic deformations or anything else the reviewer has in mind we would be happy to act upon the reviewer’s suggestions and include this in the conclusion.
>
>
> We once again thank the reviewer for their feedback - which has greatly improved our initial manuscript. We hope we have addressed all the concerns raised by the reviewer. If there is anything else we can provide - experiments, clarifications, revisions - for the reviewer to increase their rating, we would be happy to provide this.

---

### Author Response · Authors · 2025-12-02
**To the ACs: Summary of reviews and discussion**

Dear ACs,

Thank you for your engagement in this reviewing process. First, we want to emphasize: all reviews acknowledge our method as novel with improved visual quality over state-of-the-art.

More specifically, *all four reviewers to varying degrees agree that our method is the first which allows for reconstructing controllable 3DGS humans and objects, the first which allows for cross-sequence retargeting of humans interacting with novel objects* - and goes well beyond existing methods which only reconstruct 3DGS humans from multiview data. We quote: **Rw2b**: *“Experimental results demonstrate that the proposed method achieves SOTA performance.”*; **zCUP**: *"Addresses controllable human–object interactions instead of human-only avatars, enabling cross-sequence retargeting and scene composition and demonstrates consistent quantitative gains over strong baselines and compelling qualitative compositions across multiple datasets”*; **H4ie**: *“The paper simultaneously reconstructs humans and interacting objects under occlusion, advancing beyond human-only Gaussian methods”*; **ttja**: *“While the baseline methods cannot accurately reconstruct objects, the proposed method demonstrates much better visual performance than baselines”*.

Most important concerns raised by reviewers were about writing, presentation and baselines, summarized below:

**Rw2b** : 2 comments in the reviewer’s statements indicate some misunderstanding about our paper. 1) The reviewer writes *“The paper includes only one quantitative comparison with previous methods, which is insufficient to validate the results.”* Please note this is not correct. We provide quantitative comparisons on two datasets in Table 1 and 2 with *5 other SoTA methods* (and not just one as the reviewer indicates), across 3 metrics. This is acknowledged by **zCUP**: *“Demonstrates consistent quantitative gains over strong baselines and compelling qualitative compositions across multiple datasets“* and by **H4ie**: *"Experiments on the BEHAVE, NeuralDome, and DNA-Rendering datasets demonstrate high-quality reconstructions under occlusion and the ability to synthesize novel, controllable interactions"* . 2) The reviewer also writes *“The paper lacks a discussion of the proposed method’s limitations”*. Please note this is again not correct. We explicitly mention limitations in the conclusion. Our claim is backed by **H4ie** who quotes our conclusion in their writing: *“The method makes a strong assumption that it only models "one dynamic object" and that this object's motion "can be explained using only a rigid transform"”*. The reviewer has other concerns about writing and some about missing details. We have addressed each and every concern listed by the reviewer in the revised manuscript and supplementary.


**ttja**: The reviewer states: *“the application of reconstructing human interact with novel objects seems fancy, but it can be achieved with Gaussian editing or segmentation baselines”*, giving 7 papers as references. We believe this claim is without merit. These papers address the problem of reconstructing and manipulating small objects in static 3DGS scenes and in fact have no relation at all to 3D humans; integrating them in a human-object interaction pipeline is very far from being straightforward, if feasible at all. We had asked the reviewer for further evidence to support this claim in their review but the discussion period got curtailed.
In our setup the scene from which the human and object are reconstructed is dynamic and the object is most of the time heavily occluded, such that classical naive reconstruction of objects fails. Our novel object reconstruction pipeline addresses this problem with innovative technical solutions. Cited papers do not address this, assume a static scene, have no relation to 3D humans and thus are not plausible baselines for our method. We have provided comparisons with five strongest possible (open source) baselines for human, or modified human+object reconstruction and have demonstrated that our method yields consistent quantitative gains over them all. (Acknowledged by **zCUP**: *“Demonstrates consistent quantitative gains over strong baselines"*).

Reviewers mention some concerns about presentation such as color-coding of tables, inconsistent bold font; we have revised the manuscript to incorporate all the points made about presentation by the reviewers.

In the absence of reviewer engagement over the curtailed discussion period, we would really hope that our response which addresses these original concerns will be taken into account.
We believe we have addressed most of the reviewers' concerns and have improved the manuscript thanks to their help. We hope these improvements and clarifications provide a strong basis for acceptance.

---

### Meta-Review · Area_Chair_ytJd · 2026-01-06

**Summary:**

The paper proposes to reconstruct animatable human and object from multi-view posed videos. It then enables controllable rendering of novel human-object interactions under different poses and novel viewpoints. The paper initially received two rating 6 and two rating 4. The reviewers achknowledged 1) it advanced beyond human-only Gaussian methods; 2) introduced contact constraints improves physical plausibility and reduces interpenetration during animation; 3) the performance gains over prior baselines. The main concerns are 1) restrict to only one dynamic object; 2) writing quality is poor; 3) it is more engineering than research; 4) the experimental results still present noticeable artifacts, e.g. human's hand and table are not in contact, or passes through the object. The rebuttal has clarified most of the concerns and the writing is also improved with a revised version. However, by considering the proposed method's engineering-heavy characteristic and limited performance, the AC could not recommend its acceptance.

**Reviewer Concerns:**

Most of concerns have been clarified.

**Reviewer Scores:**

Reviewer ttja, who initially gives a rating 4, mentioned he would maintain his score after initial discussion. The raised concerns from Rw2b (rating 4) have been clarified. However, the AC is not sure if he would upgrade his score after reading other reviewers comment.

---

### Decision · Program_Chairs · 2026-01-26

Reject